# Developmental Exposure to Endocrine Disrupter DDT Interferes with Age-Related Involution of Thymus

**DOI:** 10.3390/ijms23126678

**Published:** 2022-06-15

**Authors:** Nataliya V. Yaglova, Sergey S. Obernikhin, Elina S. Tsomartova, Valentin V. Yaglov, Svetlana V. Nazimova, Dibakhan A. Tsomartova, Ekaterina P. Timokhina, Elizaveta V. Chereshneva, Marina Y. Ivanova, Olga V. Payushina

**Affiliations:** 1Laboratory of Endocrine System Development, Research Institute of Human Morphology of Federal State Budgetary Scientific Institution “Petrovsky National Research Centre of Surgery”, 119991 Moscow, Russia; ober@mail.ru (S.S.O.); tselso@yandex.ru (E.S.T.); vyaglov@mail.ru (V.V.Y.); pimka60@list.ru (S.V.N.); dtsomartova@mail.ru (D.A.T.); rodich@mail.ru (E.P.T.); 2Department of Histology, Cytology, and Embryology, Federal State Funded Educational Institution of Higher Education, I.M. Sechenov First Moscow State Medical University, 119435 Moscow, Russia; yelizaveta.new@mail.ru (E.V.C.); ivanova_m_y@mail.ru (M.Y.I.); payushina@mail.ru (O.V.P.)

**Keywords:** thymus, endocrine-disrupting chemicals, involution, lymphocytes, DDT

## Abstract

The impact of endocrine-disrupting chemicals on the development and involution of the immune system is a possible reason for the increased incidence of disorders associated with inappropriate immune function. The thymus is a lymphoid and also an endocrine organ, and, accordingly, its development and functioning may be impaired by endocrine disruptors. The aim was to evaluate age-related thymus involution in mature rats exposed to the endocrine disruptor DDT during prenatal and postnatal ontogeny. Methodology included in vivo experiment on male Wistar rats exposed to low doses of DDT during prenatal and postnatal development and morphological assessment of thymic involution, including the immunohistochemical detection of proliferating thymocytes. The study was carried out at the early stage of involution. Results: DDT-exposed rats exhibited a normal anatomy, and the relative weight of the thymus was within the control ranges. Histological and immunohistochemical examinations revealed increased cellularity of the cortex and the medulla, higher content of lymphoblasts, and more intensive proliferation rate of thymocytes compared to the control. Evaluation of thymic epithelial cells revealed a higher rate of thymic corpuscles formation. Conclusion: The data obtained indicate that endocrine disrupter DDT disturbs postnatal development of the thymus. Low-dose exposure to DDT during ontogeny does not suppress growth rate but violates the developmental program of the thymus by slowing down the onset of age-related involution and maintaining high cell proliferation rate. It may result in excessive formation of thymus-dependent areas in peripheral lymphoid organs and altered immune response.

## 1. Introduction

The thymus is a primary lymphoid organ, which is essential for formation of adaptive immune response. It ensures differentiation of T cell and their migration to T-dependent areas of peripheral lymphoid organs such as lymph nodes, tonsils, spleen, Peyer’s patches [1,2]. The immune system of the mammals is known to deteriorate with advancing age. Among the organs of immune defense, the thymus is the first to undergo age-dependent decline [3]. Involution of thymus function has been shown to impair T cell-mediated immunity and maintenance of tolerance to self-antigens [4,5].

Diseases associated with inappropriate immune response are on the increase over the last few decades [6,7]. There is a growing body of evidence that anthropogenic factors contribute to an increased rate of allergic diseases, autoimmune disorders, and inadequate response to viral and bacterial infections [8,9,10]. Endocrine-disrupting chemicals represent the largest part of anthropogenic and environmental hazards, which affects immunity of both adults and infants [11,12,13,14]. It is widely accepted that the thymus comprises immune and endocrine function [15,16]. Thus, the thymus may be affected by endocrine disrupters directly. Moreover, the thymus is known to be highly sensitive to glucocorticoid and sex-steroid hormones [3,17,18]; it allows endocrine disrupters to compromise thymic function indirectly through hormonal misbalance.

The investigations have showed that most known endocrine-disrupting chemicals penetrate the placental barrier [19,20,21]. Some of them exert a dismorphogenic effect on developing fetal organs [22]. After birth, endocrine disruption may also negatively influence the program of postnatal development [11]. In our previous investigations, we revealed some changes in the thymus morphology of newly born rats, exposed in utero to low doses of dichlorodiphenyltrichloroethane (DDT) [23]. DDT is one of the most wide-spread endocrine-disrupting chemicals on the planet [24]. It persists in water and soil for at least seven decades, and is found in 99.9% of neonate, infant, and adult blood samples [25]. This suggests different deviations in the pre- and postnatal development of the endocrine system and the immune system. DDT has been shown to interfere with sex-steroid production and earlier onset of puberty, which is crucial for thymic involution [26,27]. Thymic involution is characterized by a loss of weight, reduction in the cortex/medulla ratio, decreased cortex cellularity, depletion of subcapsular lymphoblasts, and replacement of epithelial stroma by connective and adipose tissue [28,29]. The hormones of the pituitary–gonadal axis are known to regulate all the above-mentioned processes [30,31]. Since DDT is a recognized antagonist of androgen receptors [32], it may prevent androgen binding and signaling by thymic cells and possibly alter postnatal development and involution. DDT also has been found to interfere with transcriptional regulation of endocrine glands development by affection of Wnt, Oct4, and Sonic Hedgehog signaling [33,34,35]. It suggests that DDT may also compromise thymus function by disruption of postnatal morphogenesis. Since thymus involution is an evolutionarily conservative process in mammals, its onset is associated with the implementation of the morphogenetic program of the organ, and not only with hormonal control of aging. This allows us to hypothesize that endocrine disruptors can fundamentally change the process of involution as a part of developmental program both by accelerating and slowing down. The disruption of the immune system development by low-dose exposure to DDT is still obscure and requires thorough investigations. The aim of the present study was to evaluate age-related changes in the thymus of post-pubertal rats exposed to DDT during prenatal and postnatal development.

## 2. Results

### 2.1. Thymus Morphology

The thymus of the DDT-exposed matured rats had typical anatomy and histology with distinct cortex and medulla (Figure 1A,B). The relative thymus weight did not differ from the control (Figure 1C). Histological examination revealed some signs of early thymic age-related involution in the control rats, such as less dense cellularity of the cortex and more prominent stromal elements, focal lymphocyte death and tangible bodies in the cortex, and rare thymic corpuscles. The rats developmentally exposed to low doses of DDT exhibited milder age-related changes. In DDT-exposed rats, histomorphometry revealed a significantly higher cortex/medulla ratio and increased cellularity of the cortex and medulla (Figure 1D–F).

Besides higher lymphocyte density, the medulla demonstrated altered thymic epithelial cell turnover. The number of thymic corpuscles in 1 mm^2^ of medulla was double that in the control (Figure 2C). The stages of thymic corpuscles’ development also differed. In the intact rats, more than half of the corpuscles were in the second stage of development (Figure 2A,D). In the DDT-exposed rats, thymic corpuscles in the first stage of development prevailed (Figure 2B,D).

### 2.2. Proliferation of Thymic Lymphocytes

Immunohistochemical evaluation revealed significant changes in proliferation rate of thymocytes in all assessed compartments. As shown in Figure 3A,C, a compact layer of subcapsular lymphoblasts in the control rats was narrowed and focally disintegrated. In the exposed rats, subcapsular lymphoblasts represented a denser integral layer, which was 40% wider than in the control group (Figure 3B,C). In addition to the subcapsular region, the inner cortex also displayed mitotically active lymphocytes, but they were diffusely located in the cortex. The percentage of Ki-67-positive lymphocytes in 1 mm^2^ of the cortex in the DDT-exposed rats was significantly higher than in the control (Figure 3D).

Immunohistochemical detection also revealed Ki-67-positive thymocytes in the medulla. In the control rats, proliferating lymphocytes were scattered in the medulla (Figure 4A). Their percentage was twice less than in the cortex. Unlike the control, the DDT-exposed rats showed a 1.5-times higher percentage of dividing lymphocytes (Figure 4C). They also were diffusely distributed in the medulla (Figure 4B).

## 3. Discussion

Thymus histophysiology comprises an influx of lymphoid progenitors to the subcapsular region, where they actively proliferate concomitant differentiation of lymphoblasts to mature T cells in the inner cortex, and their emigration from the thymus. Thymus involution is characterized by progressive loss of weight and depletion of cortex due to diminished influx of progenitor cells from the bone marrow, and a reduction in cell proliferation and differentiation [28,29]. In rats, involution of the thymus begins from the age of six weeks [36]. In the present study, we examined 10-week-old rats and found early age-related changes in the thymus of the control group. Immunohistochemical evaluation of cell proliferation revealed depletion of subcapsular layer and low mitotic activity of cortical and medullary lymphocytes indicative of extinction of thymopoiesis. DDT-exposed rats exhibited similar thymus size and no differences in the rate of cortex and medulla development. This clearly demonstrates that the endocrine disrupter did not attenuate thymus growth. Higher cellularity of the cortex and medulla reflect more active T cell production. A wider layer of lymphoblasts under the capsule and a higher percentage of dividing lymphocytes in the inner cortex provide evidence that thymopoiesis in prenatally and postnatally DDT-exposed rats does not start regressing after puberty. T cell production in the thymus is controlled by so-called thymic crosstalk, which includes initiation of progenitor cell proliferation, differentiation, and migration by thymic epithelial cells as well as induction of thymic epithelial cell differentiation and function by thymocytes [37,38]. In the present investigation, we observed a higher number of thymic corpuscles in the DDT-exposed rats. Moreover, most corpuscles were in the initial stage of development indicative of their enhanced formation. The possible role of thymic corpuscles in T cell development is still an open question [39,40,41], but developmental investigations of the rat thymus show that thymic corpuscles appear when thymopoiesis is already established, and their number decreases with age concomitantly with the regression of T cell output [42]. In our investigation, we revealed more intensive lymphocyte production and thymic epithelial cell cycling. Thus, it appears that the thymus of rats developmentally exposed to the endocrine disruptor DDT was more juvenile compared to the age control. In our previous research, we investigated the development of the proliferative response of thymic lymphocytes to T-cell mitogen Concanavalin A in rats with the same regimen of prenatal and postnatal exposure to DDT and found that the response in prepubertal and pubertal rats was adequate to age control, but after puberty—at the age of 10 weeks—it showed a significant decrease [43]. An insufficient proliferation response to mitogen is known to indicate functional immaturity of lymphocytes typical for the neonatal period of ontogeny [44]. Additionally, low proliferative response is observed in old age and associated with cell senescence and reduced immune cell renewal [45,46]. The results of the present study shed light on the cause of the insufficient proliferative response of lymphocytes. Functional insufficiency of thymic lymphocytes most likely results from higher content of low-differentiated proliferating cells incapable of blast transformation. Our findings provide evidence that developmental exposure to DDT disrupts postnatal growth of the thymus and functional maturation of thymic lymphocytes. Thus, the well-developed, functionally active appearance of the thymus cannot be considered a marker of sufficient T cell function in a case of development under persistent exposure to endocrine-disrupting chemicals. DDT has been found to disrupt postnatal development of the male reproductive system and the adrenal glands [35,47,48,49,50]. Our results show that DDT has a dysmorphorogenetic effect on the central organs of immunity along with the endocrine glands.

## 4. Materials and Methods

### 4.1. Animals

Female and male Wistar rats were obtained from Scientific Center of Biomedical Technologies of Federal Medical and Biological Agency of Russia. The rats were housed at +22–23 °C and given a pelleted standard chow ad libitum. The investigation was performed in accordance with the handling standards and rules of laboratory animals as consistent with “International Guidelines for Biomedical Researches with Animals” (1985), laboratory routine standards in the Russian Federation (Order of Ministry of Healthcare of the Russian Federation dated 19 June 2003 No.267) and “Animal Cruelty Protection Act” dated 1 December 1999, regulations of experimental animal operation approved by Order of Ministry of Healthcare of USSR No.577 dated 12 August 1977. Animal procedures were approved by the ethics committee of the Research Institute of Human Morphology (protocol N 28(4), 28 October 2021).

### 4.2. Experimental Design

The female rats weighed 180–220 g and received a solution of o,p-DDT 20 µg/L (“Sigma-Aldrich”, St. Louis, MO, USA) ad libitum instead of tap water since mating during pregnancy and lactation. After weaning, the progeny of the rat dams received the same solution of o,p-DDT during postnatal development. The progeny of intact female rats were used as a control. Only male offspring were enrolled in the experiment (10 DDT-exposed and 10 control rats). The rats were sacrificed by zoletil overdosage in the post-pubertal period at the age of 10 weeks. The average daily intake of DDT after weaning was 2.90 ± 0.12 µg/kg bw, which corresponded to DDT consumption by humans with food products with consideration for differences in DDT metabolism in rats and humans [51]. The absence of DDT, its metabolites, and related organochlorine compounds in tap water and chow was confirmed by gas chromatography in Moscow Federal Budgetary Institution of Public Health.

### 4.3. Thymus Morphology

The thymus was weighed immediately after removal and then fixed in Bouen solution. After standard histological processing, the tissue samples were embedded in paraffin. Histological sections of the thymus were stained with hematoxylin and eosin. Histological examination was performed with “Leica DM2500” light microscope (Leica Microsystems Gmbh, Wetzlar, Germany).

Computer histomorphometry of light microscope images was carried out using “ImageScope” software (Leica Microsystems Gmbh, Wetzlar, Germany). The surface area of the cortex and medulla, the number of lymphocytes in 1µm^2^ of the cortex and the medulla, and the number of thymic corpuscles in 1 µm^2^ of the medulla were measured. The stages of thymic corpuscles development were assessed according to the following classification: 1st stage—convergence of several epithelial reticular cells with higher oxyphilic cytoplasm; 2nd stage—concentric arrangement of epithelial reticular cells and accumulation of amorphous acidophilic material in the corpuscle; 3rd stage—formation of a cyst in the center of a corpuscle; 4th stage—rupture of a thymic corpuscle and elimination of debris by macrophages [52]. Cortex/medulla ratio was calculated as the ratio of the surface area of the cortex to the surface area of the medulla.

### 4.4. Immunohistochemistry

Immunohistochemical evaluation of Ki-67 was performed on paraffin-embedded tissues. After antigen retrieval with 10 mM sodium citrate (pH 6.0), endogenous peroxidase and endogenous immunoglobulins were blocked with Hydrogen Peroxide Block and Protein Block (Thermo Fisher Scientific, Waltham, MA, USA). The slides were incubated with primary antibodies to Ki-67 (1:100, Cell Marque, Rocklin, CA, USA) overnight at 8 °C. Slides processed without incubation with primary antibodies were used as a negative control. The reaction was visualized with UltraVision LP Detection System reagent kit (Thermo Fisher Scientific, Waltham, MA, USA) according to manufacturer’s recommendations. The sections were counterstained with Mayer’s hematoxylin.

Expression Ki-67 in lymphoid cells was evaluated separately in subcapsular region, inner cortex, and medulla of the thymus. In the inner cortex and the medulla rate of proliferation was assessed as a percentage of immunopositive cells with per 1 mm^2^. The width of the Ki-67-positive subcapsular layer of lymphoblasts was measured using “ImageScope” software (Leica Microsystems Gmbh, Wetzlar, Germany).

### 4.5. Statistical Analysis

The statistical analyses were carried out using the software package Statistica 7.0 (StatSoft, Tulsa, OK, USA). The central tendency and dispersion of quantitative traits with approximately normal distribution were presented as the mean and standard error of the mean (M ± SEM). Quantitative comparisons of independent groups were performed using Student’s *t*-test, taking into account the values of Levene’s test for the equality of variances. Quantitative comparisons were performed using Chi-square. Differences were considered statistically significant at *p* < 0.05.

## 5. Conclusions

The present investigation clearly demonstrates that endocrine disrupter DDT disturbs the postnatal development of the thymus. Low-dose exposure to DDT during ontogeny does not suppress growth rate but does violate the developmental program of the thymus by slowing down the onset of age-related involution and maintaining high cell proliferation rate. This may result in an excessive formation of thymus-dependent areas in the peripheral lymphoid organs and an altered immune response.

## Figures and Tables

**Figure 1 ijms-23-06678-f001:**
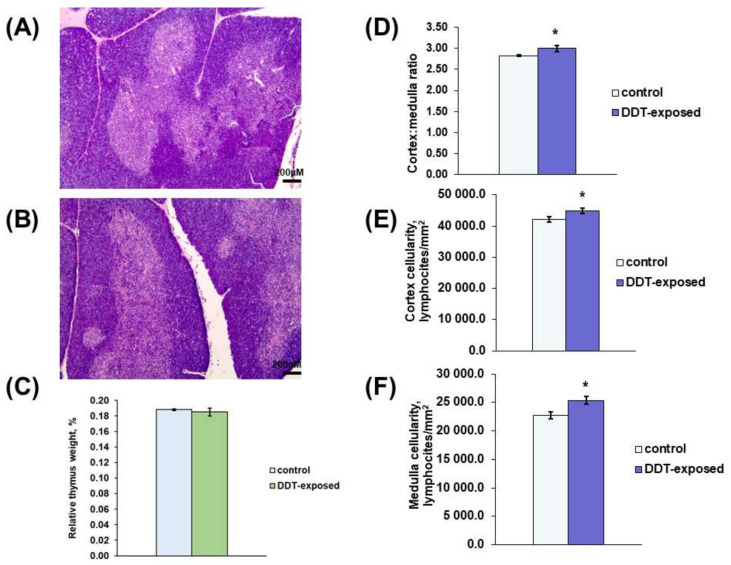
Morphology of the thymus of prenatally and postnatally exposed to DDT and intact adult rats aged 10 weeks. Histology of the thymus of the control (**A**) and DDT-exposed (**B**) rats. Magnification 50, scale bar 200 µm. Relative thymus weight (**C**), cortex/medulla ratio (**D**), lymphocyte density of the cortex (**E**) and the medulla (**F**). Data are shown as mean ± S.E.M.; *p* < 0.05 compared to the control (*).

**Figure 2 ijms-23-06678-f002:**
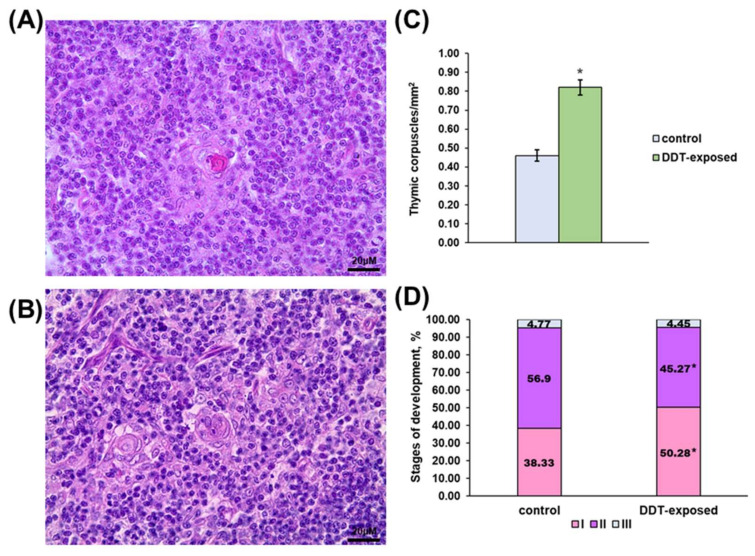
Characterization of thymic corpuscles in prenatally and postnatally exposed to DDT and intact adult rats aged 10 weeks. Structure of thymic corpuscles in the control (**A**) and DDT-exposed (**B**) rats. Magnification 400. Number of thymic corpuscles in 1 µm^2^ of the medulla (**C**). Data presents as mean ± S.E.M. Stages of thymic corpuscles development (**D**). *p* < 0.05 compared to the control (*).

**Figure 3 ijms-23-06678-f003:**
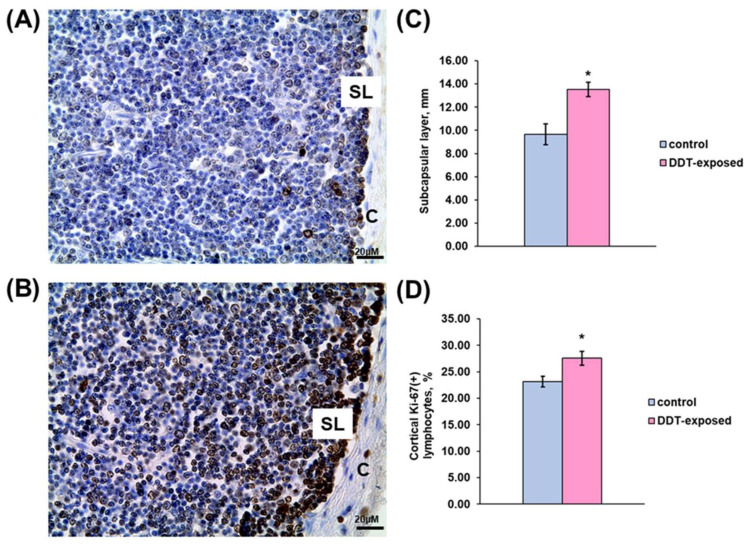
Immunohistochemical evaluation of Ki-67-positive cortical lymphocytes in prenatally and postnatally exposed to DDT and intact adult rats aged 10 weeks. Distribution of proliferating thymocytes in the control (**A**) and DDT-exposed (**B**) rats. C—capsule, SL—subcapsular lymphoblasts. Magnification 400, scale bar 20 µm. Width of subcapsular layer of mitotically active lymphoblasts (**C**), percentage of Ki-67-positive thymocytes in the cortex (**D**). Data presents as mean ± S.E.M. *p* < 0.05 compared to the control (*).

**Figure 4 ijms-23-06678-f004:**
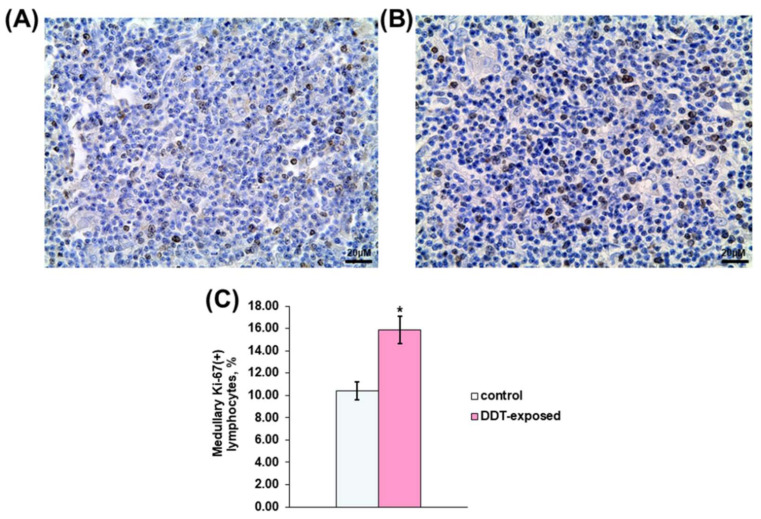
Immunohistochemical evaluation of Ki-67-positive medullary lymphocytes in prenatally and postnatally exposed to DDT and intact adult rats aged 10 weeks. Distribution of proliferating thymocytes in the control (**A**) and DDT-exposed (**B**) rats. Magnification 400, scale bar 20 µm. Percentage of Ki-67-positive thymocytes in the cortex (**C**). Data presents as mean ± S.E.M. *p* < 0.05 compared to the control (*).

## Data Availability

The data presented in this study are available from the corresponding author upon reasonable request.

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
