# Peer review of "Developmental Exposure to Endocrine Disrupter DDT Interferes with Age-Related Involution of Thymus"

_ijms, 2022, doi:10.3390/ijms23126678_

Round 1

Reviewer 1 Report

The impact of endocrine disrupting chemicals to development and involution of immune system is a possible reason for increased incidence of disorders associated with inappropriate immune function. Therefore, In this study, the age-related thymus involution in mature male Wistar rats exposed to endocrine disruptor DDT during prenatal and postnatal ontogeny was investigated by Yaglova. et al.

In the whole, this is a relatively complete study, so I recommend that this manuscript can be accepted after minor modifications.

The experimental method should be placed in the second section.

Why do the authors think that rats exposed to low-dose DDT during development show mild age-related changes in Figure 1? Are there any references?

When analyzing the experimental results, the authors should compare some previous studies and add relevant references.

Reviewer 2 Report

Dear Authors,

Thanks,

Please;

Title of the study

“Developmental exposure to endocrine disrupter DDT interferes 2 with age-related involution of thymus”

The title summarizes the main idea or ideas of your study.

A good title contains the fewest possible words that adequately describe the contents and/or purpose of your research paper.

Abstract

Please, insert: aim, methodology, results, conclusions, practical aplications and limitations.

Keywords: only 5, please

Introduction

“The investigations have showed that most known endocrine disrupting chemicals 51 penetrate placental barrier [19].”

Please, add recent studies (2021, 2022).

“It suggests that DDT may also compromise thymus 69 function by disruption of postnatal morphogenesis. In the present investigation we aimed 70 at evaluation of age-related changes in the thymus of postubertal rats exposed to DDT 71 during prenatal developmen” --------------Ok. Please, then, insert the aim of the study and:

- What is a hypothesis? A hypothesis states your predictions about what your research will find

Please, first:

Introduction

Materials and Methods

“Animals - Female and male Wistar rats were obtained from Scientific Center of Biomedical Technologies of Federal Medical and Biological Agency of Russia.”

(n= ????)

Then:

Results

Discussion

Please, insert recent studies (2021, 2022)...

Conclusion

Practical Implications

Limitations

4.5. Statistical analysis

Please, insert------------------ IBM SPSS Statistiscs, version xxxx,

Differences were considered statistically significant at p<0.05.

Effect size?

5. Conclusions

The conclusion is intended to help the reader understand why your research should matter to them after they have finished reading the paper. A conclusion is not merely a summary of your points or a re-statement of your research problem.

See:

The present investigation clearly demonstrates that endocrine disrupter DDT disturbs postnatal development of the thymus. Low-dose exposure to DDT during ontogeny does not suppress growth rate, but violates developmental program of the thymus by slowing down the onset of age-related involution and maintaining high cell proliferation  rate. It may result in excessive formation of thymus-dependent areas in peripheral lymphoid organs and altered immune response.

References

Please, insert recent studies (2021, 2022)...

Kind Regards

Round 2

Reviewer 2 Report

Dear Authors,

Thanks!

Kind Regards